# TAMING MODE COLLAPSE IN SCORE DISTILLATION FOR TEXT-TO-3D GENERATION

## ABSTRACT

Despite the remarkable performance of score distillation in text-to-3D generation, such techniques notoriously suffer from view inconsistency issues, also known as "Janus" artifact, where the generated objects fake each view with multiple front faces. Although empirically effective methods have approached this problem via time re-scheduling or prompt engineering, a statistical view to explain and tackle this problem remains elusive. In this paper, we reveal that the existing score distillation-based text-to-3D generation frameworks degenerate to maximal likelihood seeking on each view independently and thus suffer from the mode collapse problem, manifesting as the Janus artifact in practice. To tame mode collapse, we improve score distillation by re-establishing the entropy term in the correponding variational objective and derive a new update rule for 3D score distillation, dubbed *Entropic Score Distillation (ESD)*. The entropy is applied to the distribution of rendered images. Maximizing the entropy encourages diversity among different views in generated 3D assets, thereby alleviating the Janus problem. We conduct experiments with our proposed ESD, and validate that ESD can be an effective treatment for Janus artifacts in score distillation.

## 1 INTRODUCTION

Recent advancements in text-to-3D technology have attracted considerable attention, particularly for its pivotal role in automating high-quality 3D content. This is especially crucial in fields such as virtual reality and gaming, where 3D content forms the bedrock. While numerous techniques are available, the prevailing text-to-3D approach is based on score distillation, popularized by DreamFusion (Poole et al., 2022) and Wang et al. (2023a); Lin et al. (2023); Chen et al. (2023a); Tsalicoglou et al. (2023); Metzer et al. (2023); Wang et al. (2023b); Huang et al. (2023).

Score distillation involves parameterizing the 3D objective as a learnable representation, such as neural radiance fields (NeRF) (Mildenhall et al., 2020). Additionally, a 2D prior, usually a diffusion model trained on large scale 2D dataset, is utilized to optimize the fidelity of each random view of the 3D scene. Despite the notable progress achieved with score distillation-based approaches, exemplified in Poole et al. (2022); Wang et al. (2023a); Chen et al. (2023a); Lin et al. (2023); Wang et al. (2023b), it is widely observed that 3D content generated using score distillation suffers from a *Janus* effect.

To understand this drawback of score distillation, we first uncover that the optimization of existing score distillation-based text-to-3D generation degenerates to a maximum likelihood objective, making it susceptible to mode collapse. This happens because the primary goal of this objective is to solely maximize the likelihood of each view independently, without considering the diversity between different views. This oversight leads to the Janus artifact in practical applications. For example, biases in the training data may result in a frontal view of a cat having a higher likelihood than the back view, as characterized by existing diffusion models.

To address the aforementioned issue, we propose *Entropic Score Distillation* (ESD). ESD introduces entropy regularization to maximize the entropy of the distribution of the rendered images, thereby enhancing the diversity of views in generated 3D assets and alleviating the Janus problem. Our derived ESD update has a simple form. In contrast to the score distillation sampling (SDS) update proposed in DreamFusion (Poole et al., 2022), our ESD updates involve the score of the rendered image distribution, serving to maximize the entropy of the rendered image distribution. Unlike the

variational score distillation (VSD) update introduced in ProlificDreamer (Wang et al., 2023b), our update differs in that it does not depend on the camera pose in the learned score function of the rendered image distribution. This subtle difference has a more profound impact, as the score function term in ProlificDreamer has a zero mean, thereby not influencing diversity. In contrast, it is non-zero in our method, leading to an effect that enhances diversity among different views.

In practice, we find it challenging to optimize the score of the rendered image distribution without conditioning on the camera pose. To facilitate training, we discover that the gradient from the entropy can be decomposed into a combination of scores: one depends on the camera pose, and the other independent of it, with a coefficient interacting between these two terms. Such a computational paradigm can be easily implemented by classifier guidance trick where conditional and unconditional scores are trained alternatively and mixed during inference.

Through extensive experiments with our proposed ESD, we demonstrate its efficacy in alleviating the Janus problem and its significant advantages in improving 3D generation quality when compared to the baseline methods (Poole et al., 2022; Wang et al., 2023b). Since our approach is orthogonal to other approaches to relieving Janus problem (Hong et al., 2023; Armandpour et al., 2023; Huang et al., 2023), we also verify the effectiveness of our method cooperating with time scheduling prior.

## 2 BACKGROUND

### 2.1 DIFFUSION MODELS

Diffusion models, as demonstrated by various works (Sohl-Dickstein et al., 2015; Ho et al., 2020; Song & Ermon, 2019; Song et al., 2020c), have proven to be highly effective in text-to-image generation. Technically, a diffusion model learns to gradually transform a prior distribution $\mathcal{N}(\mathbf{0}, \boldsymbol{I})$ to the target distribution $p_{data}(\boldsymbol{x}|\boldsymbol{y})$ where $\boldsymbol{y}$ denotes the text prompt embeddings. The sampling trajectory is determined by a forward process with the conditional probability $p_t(\boldsymbol{x}_t|\boldsymbol{x}_0) = \mathcal{N}(\boldsymbol{x}_t|\alpha_t\boldsymbol{x}_0, \sigma_t^2\boldsymbol{I})$, where $\boldsymbol{x}_t \in \mathbb{R}^D$ represents the sample at time $t \in [0, T]$, and $\alpha_t, \sigma_t > 0$ are time-dependent diffusion coefficients. Consequently, the distribution at time $t$ can be formulated as $p_t(\boldsymbol{x}_t|\boldsymbol{y}) = \int p_{data}(\boldsymbol{x}_0|\boldsymbol{y})\mathcal{N}(\boldsymbol{x}_t|\alpha_t\boldsymbol{x}_0, \sigma_t^2\boldsymbol{I})d\boldsymbol{x}_0$. Diffusion models generate samples through a reverse process starting from Gaussian noises, which can be described by the ODE: $d\boldsymbol{x}_t/dt = -\nabla_{\boldsymbol{x}} \log p_t(\boldsymbol{x}_t)$ with the boundary condition $\boldsymbol{x}_T \sim \mathcal{N}(\mathbf{0}, \boldsymbol{I})$ (Song et al., 2020c;a; Liu et al., 2023b). Such a process requires the computation of *score function* $\nabla_{\boldsymbol{x}} \log p_t(\boldsymbol{x}_t)$ which is often obtained by fitting a time-conditioned noise estimator $\boldsymbol{\epsilon}_{\boldsymbol{\phi}} : \mathbb{R}^D \to \mathbb{R}^D$ using score matching loss (Hyvärinen & Dayan, 2005; Vincent, 2011; Song et al., 2020b).

### 2.2 TEXT-TO-3D SCORE DISTILLATION

Score distillation based 3D asset generation requires representing 3D scenes as learnable parameters $\boldsymbol{\theta} \in \mathbb{R}^N$ equipped with a differentiable renderer $g(\boldsymbol{\theta}, \boldsymbol{c}) : \mathbb{R}^N \to \mathbb{R}^D$ that projects 3D scene $\boldsymbol{\theta}$ into images with respect to the camera pose $\boldsymbol{c}$. Here $N, D$ are the dimensions of the 3D parameter space and rendered images, respectively. Neural radiance fields (NeRF) (Mildenhall et al., 2020) are often employed as the underlying 3D representation for its capability of modeling complex scenes.

Recent works by (Poole et al., 2022; Wang et al., 2023a; Lin et al., 2023; Chen et al., 2023a; Tsalicoglou et al., 2023; Metzer et al., 2023; Wang et al., 2023b; Huang et al., 2023) demonstrate the feasibility of using a pretrained 2D diffusion model to guide 3D object creation. Below, we elaborate on two score distillation schemes, *Score Distillation Sampling* (SDS) (Poole et al., 2022) and *Variational Score Distillation* (VSD) (Wang et al., 2023b), which are widely adopted.

**Score Distillation Sampling (SDS)**    SDS updates the 3D parameter $\boldsymbol{\theta}$ as follows:

$$\nabla_{\boldsymbol{\theta}}^{SDS} = \mathbb{E}_{t,\boldsymbol{c},\boldsymbol{\epsilon}\sim\mathcal{N}(\mathbf{0},\boldsymbol{I})}\left[\omega(t)\frac{\partial g(\boldsymbol{\theta}, \boldsymbol{c})}{\partial \boldsymbol{\theta}}\left(\sigma_t\nabla \log p_t(\boldsymbol{x}_t|\boldsymbol{y}) - \boldsymbol{\epsilon}\right)\right]. \quad (1)$$

Here is $\nabla \log p$ is a pretrained diffusion model and $\boldsymbol{x}_t$ is a noisy version for the rendered given by camera pose $\boldsymbol{c}$, with $\boldsymbol{x}_t = \alpha_t g(\boldsymbol{\theta}, \boldsymbol{c}) + \sigma_t\boldsymbol{\epsilon}$. Time step $t$ is sampled from a uniform distribution $\mathcal{U}[0, T]$. In practice, $\nabla \log p_t$ is often estimated with a noise estimator $\boldsymbol{\epsilon}_{\boldsymbol{\phi}^*}(\boldsymbol{x}, t, \boldsymbol{y})$ trained with denoising score matching. Meanwhile, updating $\theta$ as in Eq. equation 1 has been shown to minimize the evidence lower bound (ELBO) for the rendered images, see Wang et al. (2023a); Xu et al. (2022).

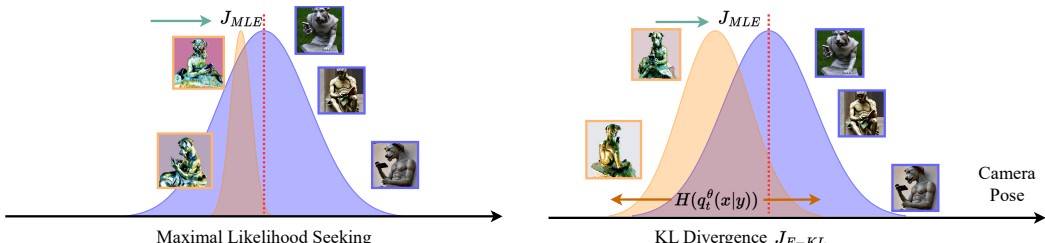

Figure 1: **Illustration of the effect of entropy regularization.**, Pure maximal likelihood seeking is opt to mode collapse. Adding entropy regularization can expand the support of fitted distribution with mode-covering behaviour.

**Variational Score Distillation (VSD)**   VSD is introduced in ProlificDreamer (Wang et al., 2023b), VSD improves upon SDS by deriving the following Wasserstein gradient flow Villani et al. (2009):

$$\nabla_{\boldsymbol{\theta}}^{VSD} = \mathbb{E}_{t,\boldsymbol{c},\boldsymbol{\epsilon}\sim\mathcal{N}(\boldsymbol{0},\boldsymbol{I})} \left[\omega(t)\frac{\partial g(\boldsymbol{\theta},\boldsymbol{c})}{\partial\boldsymbol{\theta}}\left(\sigma_t\nabla\log p_t(\boldsymbol{x}|\boldsymbol{y}) - \sigma_t\nabla\log q_t(\boldsymbol{x}|\boldsymbol{c})\right)\right]. \tag{2}$$

Similarly, $\boldsymbol{x} = \alpha_t g(\boldsymbol{\theta},\boldsymbol{c}) + \sigma_t\boldsymbol{\epsilon}$ is the noisy observation of the rendered image. In contrast to SDS, VSD introduces a new score function of the noisy rendered images conditioned on the camera pose $\mathbf{c}$. To obtain this score, Wang et al. (2023b) fine-tunes a diffusion model using images rendered from the 3D scene as follows:

$$\min_{\boldsymbol{\psi}} \mathbb{E}_{t,\boldsymbol{c},\boldsymbol{\epsilon}\sim\mathcal{N}(\boldsymbol{0},\boldsymbol{I})} \left[\omega(t)\|\boldsymbol{\epsilon}_{\boldsymbol{\psi}}(\alpha_t g(\boldsymbol{\theta},\boldsymbol{c}) + \sigma_t\boldsymbol{\epsilon}, t, \boldsymbol{c}, \boldsymbol{y}) - \boldsymbol{\epsilon}\|_2^2\right], \tag{3}$$

where $\boldsymbol{\epsilon}_{\boldsymbol{\psi}}(\boldsymbol{x}, t, \boldsymbol{c}, \boldsymbol{y})$ is the noise estimator of $\nabla\log q_t(\boldsymbol{x}|\boldsymbol{c})$ as in diffusion models. As proposed in ProlificDreamer, $\boldsymbol{\psi}$ is parameterized by LoRA (Hu et al., 2021) and initialized from a pre-trained diffusion model same as $p$.

## 3   MODE COLLAPSE IN TEXT-TO-3D SCORE DISTILLATION

Despite the remarkable performance of SDS and VSD in 3D asset generation, it is widely observed that the synthesized objects suffer from *janus* artifacts. Janus artifacts refer to the generated 3D scene containing multiple canonical views (the most representative perspective of the object such as the front face). In earlier works, Hong et al. (2023) and Huang et al. (2023) attribute this problem to unimodality of the learned 2D image distribution since the training corpus for the diffusion models are naturally biased to their canonical views per each category. In this section, we examine extant distillation schemes from a statistical view, which has been overlooked in previous literature.

In principle, natural 2D images can be seen as random projections of 3D scenes. Matching the image distribution generated by randomly sampling views from a 3D representation with a text-conditioned image distribution can recover the underlying 3D distribution. This idea got formalized by Poole et al. (2022); Wang et al. (2023b). SDS and VSD essentially correspond to the gradient of the following Kullback-Leibler (KL) divergence, i.e., $J_{SDS}(\boldsymbol{\theta}) = J_{VSD}(\boldsymbol{\theta}) = J_{KL}(\boldsymbol{\theta})$:

$$J_{KL}(\boldsymbol{\theta}) = \mathbb{E}_{t,\boldsymbol{c}} \left[\omega(t)D_{\mathrm{KL}}(q_t^{\boldsymbol{\theta}}(\boldsymbol{x}_t|\boldsymbol{c},\boldsymbol{y})\|p_t(\boldsymbol{x}_t|\boldsymbol{y}))\right], \tag{4}$$

where the expectation of $t$ is taken over $\mathcal{U}[0, T]$, $\boldsymbol{c}$ is taken over some pre-defined camera distribution $p_c$ defined on $\mathbb{SO}(3) \times \mathbb{R}^3$. Here $p_t(\boldsymbol{x}_t|\boldsymbol{y}) = \int p_0(\boldsymbol{x}_0|\boldsymbol{y})\mathcal{N}(\boldsymbol{x}_t|\alpha_t\boldsymbol{x}_0, \sigma_t^2\boldsymbol{I})d\boldsymbol{x}_0$ is the image distribution perturbed by Gaussian noises, while $q_t^{\boldsymbol{\theta}}(\boldsymbol{x}_t|\boldsymbol{c},\boldsymbol{y}) = \int q_0^{\boldsymbol{\theta}}(\boldsymbol{x}_0|\boldsymbol{c})\mathcal{N}(\boldsymbol{x}_t|\alpha_t\boldsymbol{x}_0, \sigma_t^2\boldsymbol{I})d\boldsymbol{x}_0$ models the image distribution generated by 3D parameter $\boldsymbol{\theta}$ with respect to camera pose $\boldsymbol{c}$ and diffused by Gaussian distribution. As shown by Wang et al. (2023b), $J_{KL}(\boldsymbol{\theta}) = 0$ implies $q_0^{\boldsymbol{\theta}}(\boldsymbol{x}_0|\boldsymbol{c}) = p(\boldsymbol{x}_0|\boldsymbol{y})$, i.e., the distribution of each view satisfies the text-conditioned image distribution.

However, it has not escaped from our notice that $q_0^{\boldsymbol{\theta}}(\boldsymbol{x}_0|\boldsymbol{c}) = \delta(\boldsymbol{x}_0 - g(\boldsymbol{\theta},\boldsymbol{c}))$ is a Dirac distribution for both SDS and VSD. This causes that the original KL divergence minimization (Eq. 4) degenerates

to a maximal likelihood problem:

$$J_{KL}(\boldsymbol{\theta}) = \mathbb{E}_{t,\boldsymbol{c}} \left[ \omega(t) \left( \mathbb{E}_{\boldsymbol{x_t} \sim q_t^\theta(\boldsymbol{x}_t | \boldsymbol{c}, \boldsymbol{y})} \log q_t(\boldsymbol{x}_t | \boldsymbol{c}, \boldsymbol{y}) - \mathbb{E}_{\boldsymbol{x_t} \sim q_t^\theta(\boldsymbol{x}_t | \boldsymbol{c}, \boldsymbol{y})} \log p_t(\boldsymbol{x}_t | \boldsymbol{y}) \right) \right], \quad (5)$$

$$= \underbrace{- \mathbb{E}_{t,\boldsymbol{c}} \left[ \omega(t) \, \mathbb{E}_{\boldsymbol{x_t} \sim q_t^\theta(\boldsymbol{x}_t | \boldsymbol{c}, \boldsymbol{y})} \log p_t(\boldsymbol{x}_t | \boldsymbol{y}) \right]}_{J_{MLE}(\boldsymbol{\theta})} - \underbrace{\mathbb{E}_{t,\boldsymbol{c}} \left[ \omega(t) H[q_t^{\boldsymbol{\theta}}(\boldsymbol{x}_t | \boldsymbol{c}, \boldsymbol{y})] \right]}_{const.}, \quad (6)$$

where the entropy term $H[q_t^{\boldsymbol{\theta}}(\boldsymbol{x}_t | \boldsymbol{y})]$ turns out to be a constant because $q_t^{\boldsymbol{\theta}}(\boldsymbol{x}_t | \boldsymbol{c}, \boldsymbol{y}) = \mathcal{N}(\boldsymbol{x}_t | \alpha_t g(\boldsymbol{\theta}, \boldsymbol{c}), \sigma_t^2 \boldsymbol{I})$ which has fixed entropy once $t$, $\boldsymbol{\theta}$ and $\boldsymbol{c}$ have been specified.

Eq. 6 signifies that $J_{KL}(\boldsymbol{\theta}) = J_{MLE}(\boldsymbol{\theta})$ up to a constant, hence $J_{KL}(\boldsymbol{\theta})$ shares all the minimum with $J_{MLE}(\boldsymbol{\theta})$. It is known that likelihood maximization is more prone to *mode collapse*, a phenomenon where a generative distribution fails to characterize the data diversity and concentrates on a single type of output (Goodfellow et al., 2014; Salimans et al., 2016; Metz et al., 2016; Arjovsky et al., 2017; Srivastava et al., 2017). Intuitively, minimizing $J_{MLE}(\boldsymbol{\theta})$ seeks each view *independently* to have the maximum log-likelihood on the image distribution $p(\boldsymbol{x}_0 | \boldsymbol{y})$. Since $p(\boldsymbol{x}_0 | \boldsymbol{y})$ is usually unimodal and peaks at the canonical view, each view of the scene will collapse to the same local minimum, resulting in janus artifact (Fig. 1). We postulate that the existing distillation strategies may be inherently limited by their log-likelihood seeking behaviors, which are more susceptible to biased image distributions.

## 4 SCORE DISTILLATION WITH ENTROPY REGLUARIZATION

### 4.1 ENTROPIC SCORE DISTILLATION

In this section, we highlight the importance of the entropy in score distillation. It is known that higher entropy can reflect the corresponding distribution could cover a larger support of the ambient space and thus increase the sample diversity. In Eq. 6, the entropy term is shown to diminish in the training objective, which causes each generated view to lack diversity and collapse to a single image with the highest likelihood.

To this end, we propose to bring in an entropy regularization to $J_{MLE}(\boldsymbol{\theta})$ for boosting the view diversity. Since $q_t^{\boldsymbol{\theta}}(\boldsymbol{x}_t | \boldsymbol{c}, \boldsymbol{y})$ has constant entropy, we regularize entropy for the distribution $q_t^{\boldsymbol{\theta}}(\boldsymbol{x}_t | \boldsymbol{y}) = \int q_t^{\boldsymbol{\theta}}(\boldsymbol{x}_t | \boldsymbol{c}, \boldsymbol{y}) p_c(\boldsymbol{c}) d\boldsymbol{c}$. Consider the following objective:

$$J_{Ent}(\boldsymbol{\theta}, \lambda) = - \mathbb{E}_{t,\boldsymbol{c}} \left[ \omega(t) \, \mathbb{E}_{\boldsymbol{x_t} \sim q_t^\theta(\boldsymbol{x}_t | \boldsymbol{c}, \boldsymbol{y})} \log p_t(\boldsymbol{x}_t | \boldsymbol{y}) \right] - \lambda \, \mathbb{E}_t \left[ \omega(t) H[q_t^{\boldsymbol{\theta}}(\boldsymbol{x}_t | \boldsymbol{y})] \right], \quad (7)$$

where $\lambda$ is a hyper-parameter controling the regularization strength. Intuitively, $J_{Ent}(\boldsymbol{\theta}, \lambda) = J_{MLE}(\boldsymbol{\theta}) + \lambda \, \mathbb{E}[H[q_t^{\boldsymbol{\theta}}(\boldsymbol{x}_t | \boldsymbol{y})]]$ seeks the maximal log-likelihood for each view while simultaneously enlarging the span and encouraging the diversity for the distribution $q_t^{\boldsymbol{\theta}}(\boldsymbol{x}_t | \boldsymbol{y})$ generated by randomly sampling views from the 3D parameter $\boldsymbol{\theta}$. To gain more insights, we present the following results:

**Theorem 1.** *For any $\lambda \in \mathbb{R}$ and $\boldsymbol{\theta} \in \mathbb{R}^D$, $J_{Ent}(\boldsymbol{\theta}, \lambda) = \lambda \, \mathbb{E}_t[D_{\mathrm{KL}}(q_t^{\boldsymbol{\theta}}(\boldsymbol{x}_t | \boldsymbol{y}) \| p(\boldsymbol{x}_t | \boldsymbol{y}))] + (1 - \lambda) \, \mathbb{E}_{t,\boldsymbol{c}}[D_{\mathrm{KL}}(q_t^{\boldsymbol{\theta}}(\boldsymbol{x}_t | \boldsymbol{c}, \boldsymbol{y}) \| p(\boldsymbol{x}_t | \boldsymbol{y}))] + const.$*

We prove Theorem 1 in Appendix A. Theorem 1 implies that $J_{Ent}(\boldsymbol{\theta}, \lambda)$ essentially amounts to a combination of two types of KL divergences, where the former one minimize the distribution discrepancy between $q_t^{\boldsymbol{\theta}}(\boldsymbol{x}_t | \boldsymbol{y})$ and $p_t^{\boldsymbol{\theta}}(\boldsymbol{x}_t | \boldsymbol{y})$ which marginalizes the camera pose within $q_t^{\boldsymbol{\theta}}$, while the latter is the original KL divergence $J_{KL}(\boldsymbol{\theta})$ adopted by SDS and VSD which takes expectation over $\boldsymbol{c}$ out of KL divergence.

Next, we derive the gradient of $J_{Ent}(\boldsymbol{\theta}, \lambda)$ as the update to the 3D representation:

$$\nabla_{\boldsymbol{\theta}} J_{Ent}(\boldsymbol{\theta}, \lambda) = \mathbb{E}_{t,\boldsymbol{\epsilon},\boldsymbol{c}} \left[ \omega(t) \frac{\partial g(\boldsymbol{\theta}, \boldsymbol{c})}{\partial \boldsymbol{\theta}} \left( \sigma_t \nabla_{\boldsymbol{x}} \log p_t(\boldsymbol{x}_t | \boldsymbol{y}) - \lambda \sigma_t \nabla_{\boldsymbol{x}} \log q_t^{\boldsymbol{\theta}}(\boldsymbol{x}_t | \boldsymbol{y}) \right) \right], \quad (8)$$

which can be obtained by path derivative and reparameterization trick. The full derivation is deferred to Appendix A. We name this update rule as *Entropic Score Distillation (ESD)*. Note that ESD differs from VSD as its second score function does not depend on the camera pose.

---

**Algorithm 1** ESD: Entropic score distillation for text-to-3D generation

---

**Input**: A diffusion model $\epsilon_\phi(\boldsymbol{x}, t, \boldsymbol{y})$; learnable 3D parameter $\boldsymbol{\theta}$; coefficient $\lambda$; text prompt $\boldsymbol{y}$
Initialize $\psi$ for another diffusion model $\epsilon_\psi(\boldsymbol{x}, t, \boldsymbol{y})$ with the parameter $\phi$ specified in diffusion model $\epsilon_\phi(\boldsymbol{x}, t, \boldsymbol{y})$, parameterize with LoRA
**while** not converged **do**
    Randomly sample a camera pose $\boldsymbol{c} \sim p_c$ and render a view $\boldsymbol{x}_0 = g(\boldsymbol{\theta}, \boldsymbol{c})$ from $\boldsymbol{\theta}$.
    Sample a $t \sim \mathcal{U}[0, T]$ and add Gaussian noise $\epsilon \sim \mathcal{N}(\boldsymbol{0}, \boldsymbol{I})$: $\boldsymbol{x}_t = \alpha_t \boldsymbol{x}_0 + \sigma_t \epsilon$.
    $\boldsymbol{\theta} \leftarrow \boldsymbol{\theta} - \eta \, \mathbb{E}\left[\omega(t)\frac{\partial g(\boldsymbol{\theta}, \boldsymbol{c})}{\partial \boldsymbol{\theta}}(\epsilon_\phi(\boldsymbol{x}_t, t, \boldsymbol{y}) - \lambda \epsilon_\psi(\boldsymbol{x}_t, t, \emptyset, \boldsymbol{y}) - (1-\lambda)\epsilon_\psi(\boldsymbol{x}_t, t, \boldsymbol{c}, \boldsymbol{y}))\right]$
    With probability $1 - p_\emptyset$, $\min_\psi \mathbb{E}_{t, \boldsymbol{c}, \epsilon \sim \mathcal{N}(\boldsymbol{0}, \boldsymbol{I})}\left[\omega(t)\|\epsilon_\psi(\boldsymbol{x}_t, t, \boldsymbol{c}, \boldsymbol{y}) - \epsilon\|_2^2\right]$
    Otherwise, $\min_\psi \mathbb{E}_{t, \boldsymbol{c}, \epsilon \sim \mathcal{N}(\boldsymbol{0}, \boldsymbol{I})}\left[\omega(t)\|\epsilon_\psi(\boldsymbol{x}_t, t, \emptyset, \boldsymbol{y}) - \epsilon\|_2^2\right]$.
**end while**
**Return** $\boldsymbol{\theta}$

---

## 4.2 CLASSIFIER-FREE GUIDANCE TRICK

Similar to SDS and VSD, we can approximate $\nabla_{\boldsymbol{x}} \log p_t(\boldsymbol{x}_t | \boldsymbol{y})$ via a pre-trained diffusion model $\epsilon_\phi(\boldsymbol{x}_t, t, \boldsymbol{y})$. However, $\nabla_{\boldsymbol{x}} \log q_t^{\boldsymbol{\theta}}(\boldsymbol{x} | \boldsymbol{y})$ is not readily available. We found that directly fine-tuning a pre-trained diffusion model using rendered images to approximate $\nabla_{\boldsymbol{x}} \log q_t^{\boldsymbol{\theta}}(\boldsymbol{x} | \boldsymbol{y})$, akin to Prolific-Dreamer (Wang et al., 2023b), does not yield robust performance. We conjecture this difficulty arises from the removal of the camera condition, increasing the complexity of the distribution to be fitted.

To tackle this problem, we recall the result in Theorem 1 that $J_{Ent}(\boldsymbol{\theta}, \lambda)$ can be written in terms of two KL divergence losses. Therefore, its gradient can be decomposed as a (convex) combination of their gradients, which correspond to unconditional and conditional score functions in terms of the camera pose $\boldsymbol{c}$, respectively:

$$\nabla_{\boldsymbol{\theta}} J_{Ent}(\boldsymbol{\theta}, \lambda) = \mathbb{E}\left[\omega(t)\frac{\partial g(\boldsymbol{\theta}, \boldsymbol{c})}{\partial \boldsymbol{\theta}}(\sigma_t \nabla_{\boldsymbol{x}} \log p_t(\boldsymbol{x}_t | \boldsymbol{y}) - \lambda \sigma_t \nabla_{\boldsymbol{x}} \log q_t^{\boldsymbol{\theta}}(\boldsymbol{x}_t | \boldsymbol{y})\right. \tag{9}$$

$$\left. - (1-\lambda)\sigma_t \nabla_{\boldsymbol{x}} \log q_t^{\boldsymbol{\theta}}(\boldsymbol{x}_t | \boldsymbol{c}, \boldsymbol{y}))\right].$$

With the above formulation, ESD can be implemented via the classifier-free guidance (CFG) trick, which was initially proposed to balance the variety and quality of text-conditionally generated images from diffusion models (Ho & Salimans, 2022). To be more specific, we plug in pre-trained and fine-tuned diffusion models to surrogate score functions in Eq. 10:

$$\nabla_{\boldsymbol{\theta}} J_{Ent}(\boldsymbol{\theta}, \lambda) = -\mathbb{E}\left[\omega(t)\frac{\partial g(\boldsymbol{\theta}, \boldsymbol{c})}{\partial \boldsymbol{\theta}}(\epsilon_\phi(\boldsymbol{x}_t, t, \boldsymbol{y}) - \lambda \epsilon_\psi(\boldsymbol{x}_t, t, \emptyset, \boldsymbol{y}) - (1-\lambda)\epsilon_\psi(\boldsymbol{x}_t, t, \boldsymbol{c}, \boldsymbol{y}))\right],$$

where $\emptyset$ denotes the placeholder embedding to indicate unconditional score estimation, and $\epsilon_\psi$ is the fine-tuned diffusion model using rendered images similar to Wang et al. (2023b). We follow the training strategy suggested by classifier-free guidance which takes random turns with a probability $p_\emptyset$ to balance the training of conditional and unconditional score functions. See Algorithm 1.

## 4.3 DISCUSSION

In VSD, the camera condition score plays a significant role in facilitating visual quality. Intuitively, such conditioning can equip the tuned diffusion model with multi-view priors (Liu et al., 2023a). Also, Hertz et al. (2023) suggests such a method can be useful to stabilize the update of the implicit parameters. However, ESD initiates a counter-argument that the camera condition may not be always advantageous as the resultant KL divergence leads to a likelihood maximization algorithm that is prone to mode collapse. ESD differs from VSD in that it introduces entropy regularization to enhance diversity across views, a feature absent in VSD. And the update of ESD has a simple form similar to VSD, allowing for a straightforward implementation based on VSD.

We also note that by Theorem 1, ESD also optimizes for a mode-seeking KL divergence. This suggests that ESD may still lead to mode collapse especially when the target image distribution is overly concentrated on one peak (Salimans et al., 2016). Careful tuning of $\lambda$ is also necessary to balance

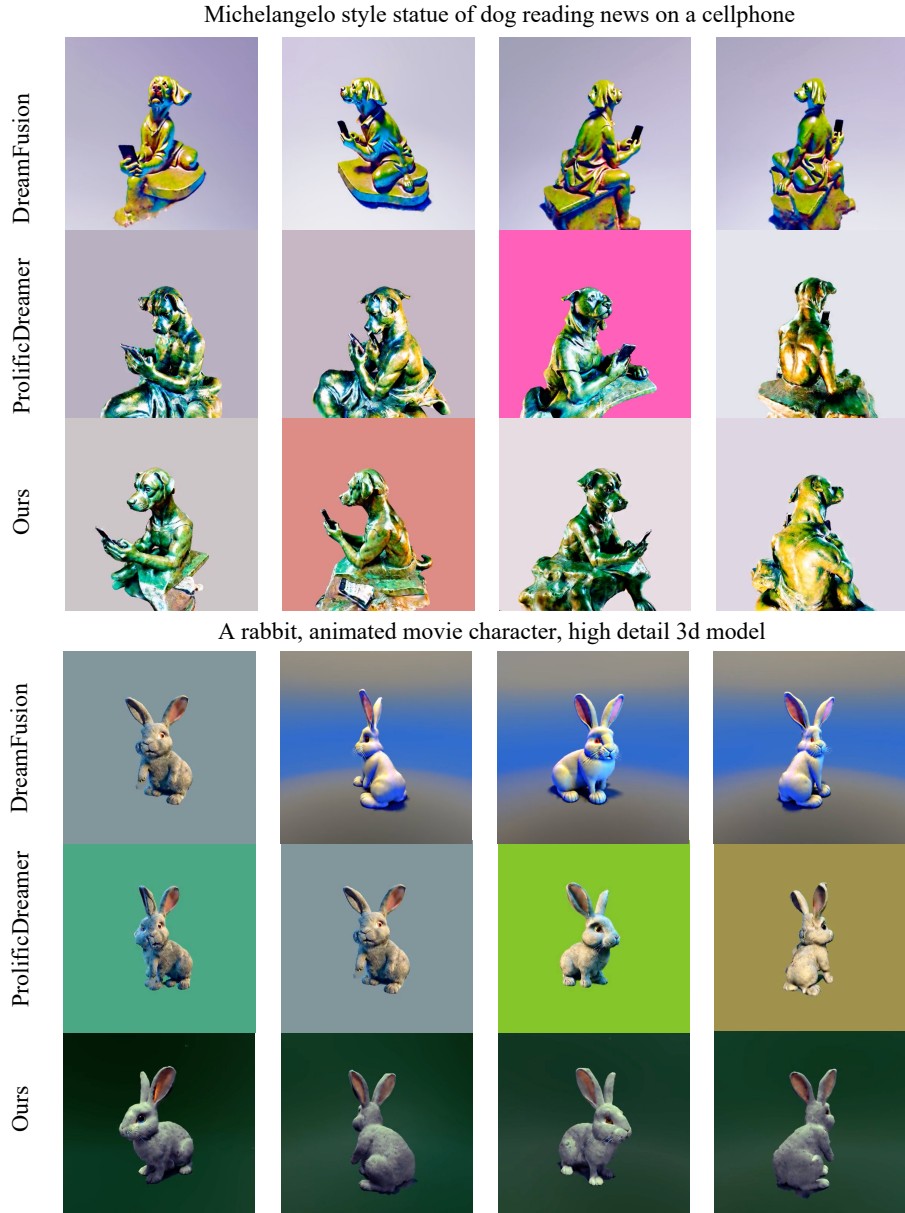

Figure 2: **Qualitative Results.** Our proposed outperforms SDS and VSD in terms of better geometry and well-constructed texture details. Our results deliver photo-realistic and diverse rendered views, while baseline methods more or less suffer from the Janus problem.

the sharpness and details for each view and diversity across views. It also remains open whether ESD can further benefit multi-particle based VSD or amortized text-to-3D training (Lorraine et al., 2023).

# 5 OTHER RELATED WORKS

## 5.1 TEXT-TO-IMAGE DIFFUSION MODEL

The denoising diffusion model (Sohl-Dickstein et al., 2015) learns to generate data through an iterative denoising process. The forward process adds Gaussian noise to clean images, while a learnable reverse process is adopted to denoise. Equipped with large-scale image-text paired datasets, many works (Rombach et al., 2022; Nichol et al., 2021; Ramesh et al., 2022; Saharia et al., 2022) scale up to tackle text-to-image generation. Among them, Stable Diffusion (Rombach et al., 2022) attracted

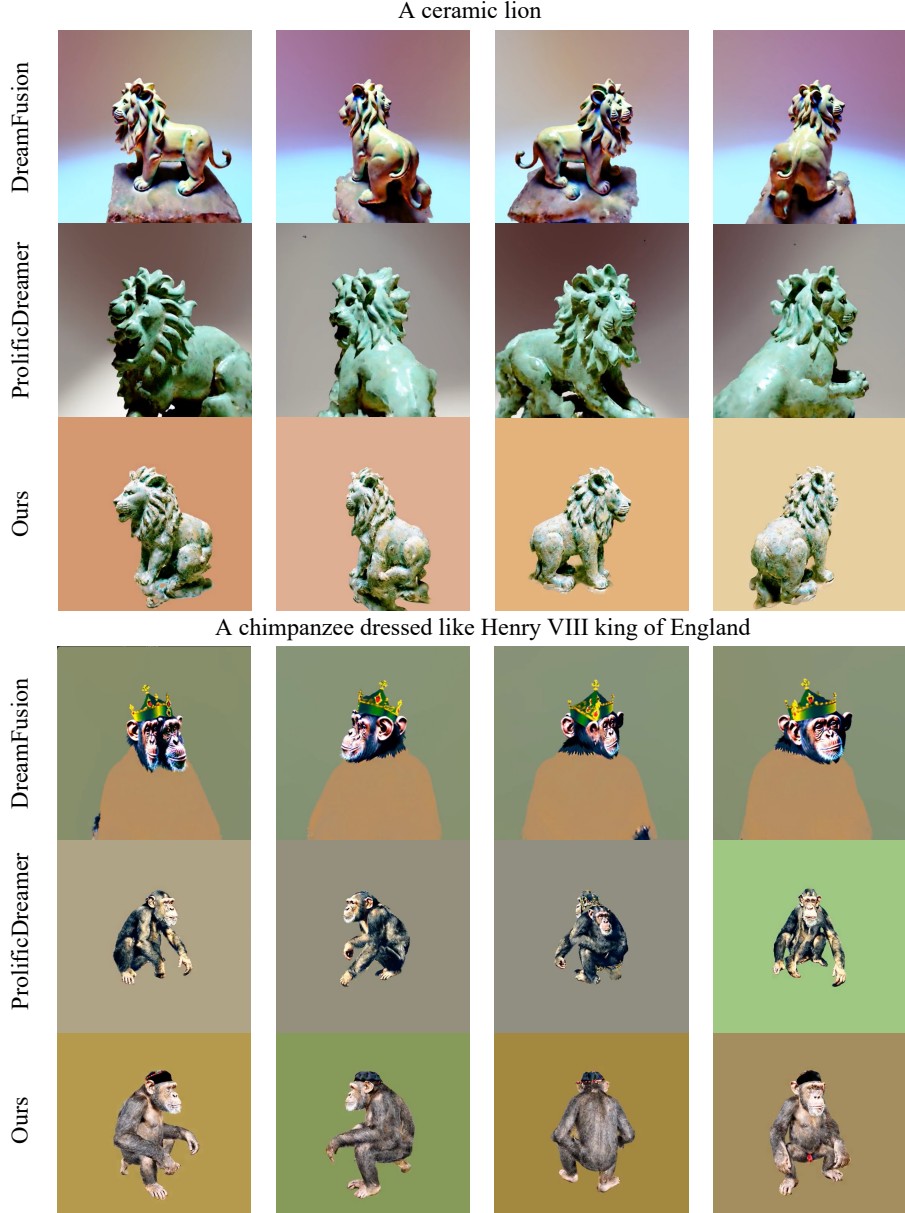

Figure 3: **Qualitative Results.** We combine our proposed ESD with timestep scheduling in DreamTime (Huang et al., 2023) and compare it against baseline methods.

great interest among the open-source community since it reduced the computation cost by diffusing in the low-resolution latent space instead of directly in the pixel space. In addition, text-to-image diffusion models have also found applications in various computer vision tasks, including text-to-3D (Poole et al., 2022; Singer et al., 2023), image-to-3D (Xu et al., 2022), text-to-svg (Jain et al., 2023), text-to-video (Singer et al., 2022; Khachatryan et al., 2023), etc.

## 5.2 3D GENERATION WITH 2D PRIORS

Well-annotated 3D data requires great effort to collect. Instead, numerous researchers study how to learn 3D generative models using 2D supervision. DreamField leverages CLIP to guide the generated images. Early attempts, including pi-GAN (Ranftl et al., 2021), EG3D (Chan et al., 2022), GRAF (Schwarz et al., 2020), GIRAFFE (Niemeyer & Geiger, 2021), adopt adversarial loss between the rendered images and natural images. More recently, with the rapid development of text-to-image diffusion models, diffusion-based image priors have attracted great interest. DreamFusion (Poole

et al., 2022) proposes score distillation sampling that effectively distills prior knowledge from Imagen (Saharia et al., 2022). SJC (Wang et al., 2023a) derives Perturb and Average Scoring from another theoretical viewpoint. ProlificDreamer (Wang et al., 2023b) proposes variational score distillation that effectively enhances the generation quality.

## 5.3 TECHNIQUES TO IMPROVE SCORE DISTILLATION

There are various approaches improving score distillation. Magic3D and Fantasia3D utilize mesh and DMTet to disentangle the optimization of geometry and texture. TextMesh and 3DFuse use depth-conditioned text-to-image diffusion priors that support geometry-aware texturing. Prompt debiasing(Hong et al., 2023) and Perp-neg (Armandpour et al., 2023) study to refine the text prompts for a better 3D generation. DreamTime (Huang et al., 2023) and RED-Diff (Mardani et al., 2023) investigate the timestep scheduling in the score distillation process. HIFA (Zhu & Zhuang, 2023) adopts multiple diffusion steps for distillation. Score distillation also works with auxiliary losses, including CLIP loss (Xu et al., 2022) and adversarial loss (Shao et al., 2023; Chen et al., 2023b).

## 6 EVALUATION METRICS

In our experiments, we consider the following metrics to numerically evaluate the generated 3D results, which can effectively quantify the similarity with the text prompts, distribution fitness, rendering quality, and view diversity.

**CLIP Distance.** We compute the average distance between rendered images and the text embedding to reflect the relevance between generated results and the specified text prompt. Specifically, we render 120 views from the generated 3D representations, and for each view, we obtain an embedding vector through the image encoder of a CLIP model (Wang et al., 2022). In the meantime, we compute the text embedding utilizing the text encoder. The CLIP distance is computed as the cosine similarity between the image embeddings and text embeddings averaged over 120 views.

**Fréchet inception distance (FID).** As shown in Sec. 3 and 4.1, score distillation essentially matches distributions via KL divergence. Hence, it is reasonable to employ FID to measure the distance between two matched distributions to quantify the effectiveness of the algorithms. Our goal is to measure the distance between generated image distribution via randomly rendering 3D parameters $q^{\boldsymbol{\theta}}(\boldsymbol{x}_0|\boldsymbol{y})$ and the text-conditioned image distribution $p(\boldsymbol{x}_0|\boldsymbol{y})$ represented by a diffusion model. Therefore, we sample 1k images with pre-trained latent diffusion model (Rombach et al., 2022) given text prompts as the ground truth image dataset, and render 120 views from the optimized 3D parameters as the generated image dataset. Then standard FID (Heusel et al., 2017) is computed between these two image corpus.

**Inception Quality and Variety.** We also utilize entropy to reflect the generated image quality and diversity inspired by Inception Score (IS) (Salimans et al., 2016). We propose Inception Quality (IQ) and Inception Variety (IV), formulated as below:

$$IQ(\boldsymbol{\theta}) = \mathbb{E}_{\boldsymbol{c}}\left[H[p(\boldsymbol{y}|g(\boldsymbol{\theta}, \boldsymbol{c}))]\right], \quad IV(\boldsymbol{\theta}) = H[\mathbb{E}_{\boldsymbol{c}}[p(\boldsymbol{y}|g(\boldsymbol{\theta}, \boldsymbol{c}))]], \quad (10)$$

where $p(\boldsymbol{y}|\boldsymbol{x})$ is a pre-trained classifier. IQ computes the average entropy of the label logits predicted for all rendered views, while IV computes the entropy of the averaged label logits of all rendered views. Intuitively, the entropy of the predicted logits suggests the confidence of the classifier, which also indicates the quality of an image. Therefore, the smaller IQ means the generated 3D assets have better visual quality. In the meanwhile, entropy also characterizes the uniformness of the distribution. The higher IV signifies that each rendered view has a distinct label prediction, meaning the 3D creation has higher view diversity. We compute IQ and IV over 120 renderings like the other two metrics.

## 7 RESULTS AND ANALYSIS

**Quantitative Comparison** With the help of the aforementioned metrics, we conduct extensive quantitative comparisons against existing methods. We present the results in Tab. 1 for four different test cases. It can be observed that in all four metrics, our proposed ESD performs as well as baseline

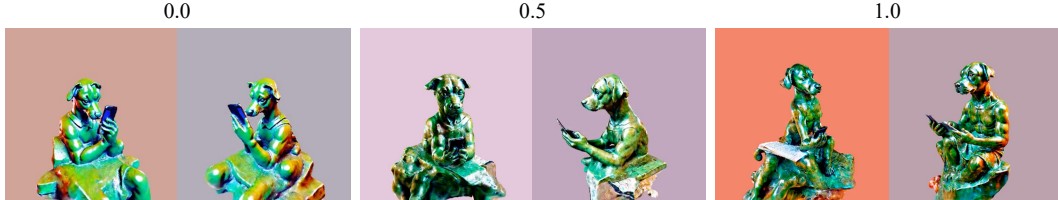

Figure 4: **Ablation studies.** We investigate the choice of different classifier-free guidance weights. Prompt: Michelangelo style statue of dog reading news on a cellphone.

Table 1: Quantitative comparisons against SDS and VSD.

Prompt: A rabbit, animated movie character, high detail 3d model

|  | CLIP ($\downarrow$) | FID ($\downarrow$) | IQ ($\downarrow$) | IV ($\uparrow$) |
|---|---|---|---|---|
| SDS | **0.712** | 200.084 | 4.365 | **4.970** |
| VSD | 0.720 | 150.120 | **1.083** | 1.173 |
| Ours | 0.725 | **149.763** | 1.567 | 1.567 |

Prompt: Michelangelo style statue of dog reading news on a cellphone

|  | CLIP ($\downarrow$) | FID ($\downarrow$) | IQ ($\downarrow$) | IV ($\uparrow$) |
|---|---|---|---|---|
| SDS | 0.694 | 365.304 | 4.469 | **5.119** |
| VSD | 0.758 | 296.168 | **2.514** | 3.041 |
| Ours | **0.685** | **292.716** | 2.523 | 4.080 |

Prompt: A plush dragon toy

|  | CLIP ($\downarrow$) | FID ($\downarrow$) | IQ ($\downarrow$) | IV ($\uparrow$) |
|---|---|---|---|---|
| SDS | 0.889 | 243.984 | 4.622 | 4.208 |
| VSD | 0.821 | 273.495 | **4.382** | 4.240 |
| Ours | **0.815** | **237.518** | 4.436 | **4.971** |

Prompt: A rotary telephone carved out of wood

|  | CLIP ($\downarrow$) | FID ($\downarrow$) | IQ ($\downarrow$) | IV ($\uparrow$) |
|---|---|---|---|---|
| SDS | 0.853 | 309.929 | 3.478 | 4.179 |
| VSD | 0.855 | 305.920 | 3.469 | 4.214 |
| Ours | **0.846** | **299.578** | **3.332** | **4.439** |

methods, if not superior to them. Our ESD achieves the best performance in terms of CLIP and FID. Not only do our generated results achieve the best alignment with the input text prompt, but they also reach the best distribution matching image quality. Our models' competitive IQ and IV results further suggest that our delivered 3D assets attain better view diversity and view quality.

**Qualitative Comparison** We present qualitative comparisons in Fig. 2 and Fig. 3. Specifically, we first conduct extensive experiments using the vanilla timestep scheduling strategy in Fig. 2. Then, we combine our proposed ESD with the time-prioritized mechanism in DreamTime (Huang et al., 2023) and showcase its ability in Fig. 3. It is clearly shown that our proposed ESD delivers better geometry with the Janus problem alleviated. In comparison, the results presented by SDS and VSD all contain more or less corrupted geometry with multi-face structures. The entropy term introduced in our ESD applies to the distribution of rendered images and encourages diversity across views. We provide additional comparisons in Fig. 5 and in video supplementary file.

**Ablation Studies** As shown in Fig. 4, we conduct ablation studies on the choice of different classifier-free guidance weights. We observe that when setting the weight to zero, the produced 3D asset contains blurry geometry and smoothed textures with fewer details. If the weight is set to one, on the other hand, the generated results will suffer from the Janus problem with multiple faces being presented. We empirically find that when setting the weight to the middle ground (*e.g.* 0.5), the final 3D asset will benefit from better details and well-constructed geometry with less Janus problem.

## 8 CONCLUSIONS

In this paper, we reveal that existing score distillation methods degenerate to maximal likelihood seeking on each view independently, leading to the mode collapse problem. We identify that re-establishing the entropy term in the corresponding variational objective brings a new update rule for better score distillation. With the help of the entropy term applied to the distribution of rendered images, our proposed Entropic Score Distillation (ESD) encourages diversity across views, mitigating the Janus problem.

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

## A    DEFERRED DERIVATION

Let us consider another KL divergence objective below.

$$\min_{\boldsymbol{\theta}} \mathbb{E}_t \left[ D_{\mathrm{KL}}(q_t(\boldsymbol{x}|\boldsymbol{\theta}) \| p_t(\boldsymbol{x}|\boldsymbol{y})) \right], \tag{11}$$

which, compared with Eq. 4, we move the expectation of $\boldsymbol{c}$ inside the KL divergence. Now we compute the gradient of objective Eq. 11:

$$\nabla_{\boldsymbol{\theta}} \mathbb{E}_t \left[ D_{\mathrm{KL}}(q_t(\boldsymbol{x}|\boldsymbol{\theta}) \| p_t(\boldsymbol{x}|\boldsymbol{y})) \right] = \mathbb{E}_t \left[ \nabla_{\boldsymbol{\theta}} D_{\mathrm{KL}}(q_t(\boldsymbol{x}|\boldsymbol{\theta}) \| p_t(\boldsymbol{x}|\boldsymbol{y})) \right] \tag{12}$$

$$= \mathbb{E}_t \left[ \nabla_{\boldsymbol{\theta}} \mathbb{E}_{\boldsymbol{x} \sim q_t(\boldsymbol{x}|\boldsymbol{\theta})} \log \frac{q_t(\boldsymbol{x}|\boldsymbol{\theta})}{p_t(\boldsymbol{x}|\boldsymbol{y})} \right] \tag{13}$$

$$= \mathbb{E}_t \left[ \nabla_{\boldsymbol{\theta}} \mathbb{E}_{\boldsymbol{\epsilon}, \boldsymbol{c}} \log \frac{q_t(g(\boldsymbol{\theta}, \boldsymbol{c}) + \boldsymbol{\epsilon})|\boldsymbol{\theta})}{p_t(g(\boldsymbol{\theta}, \boldsymbol{c}) + \boldsymbol{\epsilon})|\boldsymbol{y})} \right] \tag{14}$$

$$= \mathbb{E}_{t, \boldsymbol{\epsilon}, \boldsymbol{c}} \left[ \nabla_{\boldsymbol{\theta}} \log q_t(g(\boldsymbol{\theta}, \boldsymbol{c}) + \boldsymbol{\epsilon})|\boldsymbol{\theta}) - \nabla_{\boldsymbol{\theta}} \log p_t(g(\boldsymbol{\theta}, \boldsymbol{c}) + \boldsymbol{\epsilon})|\boldsymbol{y}) \right] \tag{15}$$

$$= \mathbb{E}_{t, \boldsymbol{\epsilon}, \boldsymbol{c}} \left[ \underbrace{\nabla_{\boldsymbol{\theta}} \log q_t(g(\boldsymbol{\theta}, \boldsymbol{c}) + \boldsymbol{\epsilon})|\boldsymbol{\theta})}_{(a)} - \frac{\partial g(\boldsymbol{\theta}, \boldsymbol{c})}{\partial \boldsymbol{\theta}} \nabla_{\boldsymbol{x}} \log p_t(g(\boldsymbol{\theta}, \boldsymbol{c}) + \boldsymbol{\epsilon})|\boldsymbol{y}) \right] \tag{16}$$

where the parameterization trick $\boldsymbol{x} = g(\boldsymbol{\theta}, \boldsymbol{c}) + \boldsymbol{\epsilon}$ is applied. Below we examine that the expectation of $(a)$ by path derivative:

$$\mathbb{E}_{t, \boldsymbol{\epsilon}, \boldsymbol{c}} \left[ \nabla_{\boldsymbol{\theta}} \log q_t(g(\boldsymbol{\theta}, \boldsymbol{c}) + \boldsymbol{\epsilon})|\boldsymbol{\theta}) \right] = \mathbb{E}_{t, \boldsymbol{\epsilon}, \boldsymbol{c}} \left[ \nabla_{\boldsymbol{\theta}} \log q_t(\boldsymbol{x}|\boldsymbol{\theta}) + \frac{\partial g(\boldsymbol{\theta}, \boldsymbol{c})}{\partial \boldsymbol{\theta}} \nabla_{\boldsymbol{x}} \log q_t(\boldsymbol{x}|\boldsymbol{\theta}) \right] \Bigg|_{\boldsymbol{x} = g(\boldsymbol{\theta}, \boldsymbol{c}) + \boldsymbol{\epsilon}} \tag{17}$$

$$= \underbrace{\mathbb{E}_{t, \boldsymbol{x} \sim q_t(\boldsymbol{x}|\boldsymbol{\theta})} \nabla_{\boldsymbol{\theta}} \log q_t(\boldsymbol{x}|\boldsymbol{\theta})}_{=0} + \mathbb{E}_{t, \boldsymbol{\epsilon}, \boldsymbol{c}} \left[ \frac{\partial g(\boldsymbol{\theta}, \boldsymbol{c})}{\partial \boldsymbol{\theta}} \nabla_{\boldsymbol{x}} \log q_t(g(\boldsymbol{\theta}, \boldsymbol{c}) + \boldsymbol{\epsilon}|\boldsymbol{\theta}) \right] \tag{18}$$

$$= \mathbb{E}_{t, \boldsymbol{\epsilon}, \boldsymbol{c}} \left[ \frac{\partial g(\boldsymbol{\theta}, \boldsymbol{c})}{\partial \boldsymbol{\theta}} \nabla_{\boldsymbol{x}} \log q_t(g(\boldsymbol{\theta}, \boldsymbol{c}) + \boldsymbol{\epsilon}|\boldsymbol{\theta}) \right] \tag{19}$$

And in all,

$$\mathbb{E}_{t, \boldsymbol{\epsilon}, \boldsymbol{c}} \left[ \frac{\partial g(\boldsymbol{\theta}, \boldsymbol{c})}{\partial \boldsymbol{\theta}} \left( \nabla_{\boldsymbol{x}} \log p_t(g(\boldsymbol{\theta}, \boldsymbol{c}) + \boldsymbol{\epsilon})|\boldsymbol{y}) - \nabla_{\boldsymbol{x}} \log q_t(g(\boldsymbol{\theta}, \boldsymbol{c}) + \boldsymbol{\epsilon}|\boldsymbol{\theta})) \right) \right] \tag{20}$$

## B    ADDITIONAL RESULTS

We place the deferred qualitative results in Fig. 5.

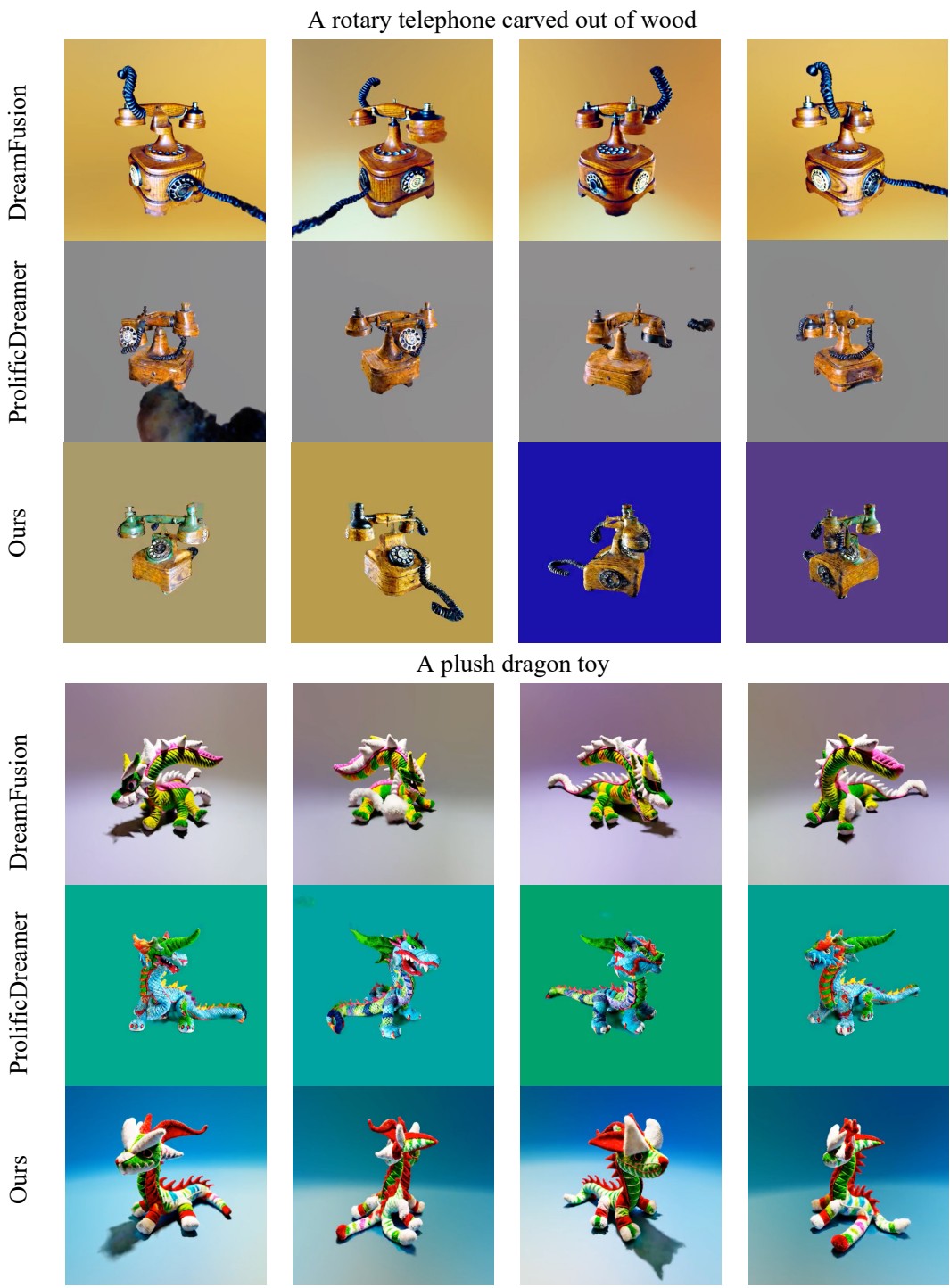

Figure 5: **Qualitative Results.** ESD outperforms SDS and VSD.

