# OpenReview forum: "Taming Mode Collapse in Score Distillation for Text-to-3D Generation"
_ICLR.cc/2024/Conference — ICLR 2024 Conference Withdrawn Submission_

### Official Review · Reviewer_S4R1 · 2023-10-30

**Soundness:** 4 excellent
**Presentation:** 3 good
**Contribution:** 2 fair
**Rating:** 6
**Confidence:** 4

**Summary:**

To tame mode collapse, this work improves score distillation by re-establishing the entropy term in the correponding variational objective and derive a new update rule for 3D score distillation, dubbed Entropic Score Distillation (ESD). The entropy is applied to
the distribution of rendered images. Maximizing the entropy encourages diversity among different views in generated 3D assets, thereby alleviating the Janus problem. Some experiments are conducted to demonstrate the effectiveness of ESD.

**Strengths:**

1. The idea is simple and intuitive.
2. The math part seems not wrong to me.
3. Experiments show the ESD performs better than baselines.

**Weaknesses:**

1. The main concern is about the evaluation. Since the results can be cherry-picked, could you report the success rate of generation, i.e., how many generated results does not have Janus problem?
2. Missing baselines: Several methods on eliminating the Janus problems should be compared:  [1] Hong, Susung, Donghoon Ahn, and Seungryong Kim. "Debiasing scores and prompts of 2d diffusion for robust text-to-3d generation." arXiv preprint arXiv:2303.15413 (2023).
[2] Armandpour, Mohammadreza, et al. "Re-imagine the Negative Prompt Algorithm: Transform 2D Diffusion into 3D, alleviate Janus problem and Beyond." arXiv preprint arXiv:2304.04968 (2023).

**Questions:**

Same as weaknesses.

---

### Official Review · Reviewer_Ybnt · 2023-11-01

**Soundness:** 2 fair
**Presentation:** 3 good
**Contribution:** 3 good
**Rating:** 5
**Confidence:** 4

**Summary:**

The paper addresses a mode collapse of Score Distllation (SDS) in text-to-3D, where generated 3D scenes often suffer from janus artifacts (such as a two-face cat). The authors propose Entropic Score Distillation (ESD) that incorporates entropy regularization to enhance view diversity and prevent mode collapse. They also introduce a classifier-free guidance trick to effectively implement ESD.

**Strengths:**

1. This paper introduces classifier-free guidance for the camera condition in VSD's gradient update rule, providing a valuable tool for text-to-3D field by demonstrating that this can be interpreted as a term considering entropy for the q distribution.
2. The motivation is intuitive, and the method is easy to implement.

**Weaknesses:**

The major concern is the lack of experiments and analysis. This paper only shows results for six prompts, including all results in the main paper, appendix, and supplementary video. I think it could be insufficient to convince that the proposed method adequately addresses the Janus problem. Although additional quantitative results are provided, they aren't metrics about the Janus problem or mode collapse, which this paper mainly addresses. I understand there's currently no proper metric that fully reflects multi-view consistency(Janus problem, mode collapse). However, it's recommended that the authors provide additional results in the appendix or conduct user study.

**Questions:**

There are a few ambiguous points I'd like to ask about.
1. At the end of page 3, q() is interpreted as a Dirac distribution in both VSD, SDS. As far as I know, in VSD, q ()is approximated as a score estimation of LoRA + Diffusion when deriving the gradient, so I thougt it is hard to see q() as a Dirac distribution in VSD, even for the case of a single particle. Could you please explain about this if there are points I missed?

2. It is mentioned that the proof of Theorem 1 is in Appendix A, but I failed to find or understand how J_ent() can derived to the two KL divergences (it's important because the two KL divergences is interpreted as CFG later). Further explanation is needed.

---

### Official Review · Reviewer_nCDX · 2023-11-07

**Soundness:** 2 fair
**Presentation:** 3 good
**Contribution:** 1 poor
**Rating:** 3
**Confidence:** 5

**Summary:**

This paper considers text-to-3D generation following the score distillation methods such as DreamFusion, ProlificDreamer, and propose a new method called Entropic Score Distillation, which incorporates entropy regularization term in score distillation sampling to evade the mode collapse issue observed from the prior methods. In specific, the author regards the problem of mode collapse arises due to the maximum likelihood estimation objective, i.e., driven from the KL divergence minimization, which is shown to be prone to the mode collapse as from various prior generative model research. To handle this challenge, the author adds entropic regularization, which can be efficiently implemented by changing the classifier-free guidance trick. They qualitatively and quantitatively show the results of 4 different cases and compare with DreamFusion and ProlificDreamer to validate their findings.

**Strengths:**

The strength of this paper is listed as follows:
- The paper includes a detailed algorithm in implementing proposed method Entropic Score Distillation (ESD), which suggests a clear path for replication and verification. Also, I believe there is no other computational burden in implementing ESD.
- The Janus problem is considered as a main problem of text-to-3D generation with 2D diffusion prior, in which the author discusses how to overcome this problem.

**Weaknesses:**

Despite the method is easy-to-follow, I believe there are many room for improvement in improving the proposed method:
- Motivation: It is unclear how the mode collapse is related to the Janus problem. In specific, what is the precise definition of mode collapse in text-to-3D generation? In the general context, I believe the term mode collapse is used for the inability of the generative model to generate diverse output in distribution-level. However, it seems like the paper considers a problem in generating a single 3D content. Thus, it will be clearer if the paper considers the mode collapse problem of certain prompts and aims to alleviate the issue, or aims to handle the widespread mode collapse problem of text-to-3D generation. In such a case, how to measure the mode collapse problem in text-to-3D generation? I think the Inception score alone cannot faithfully correlate to the mode collapse problem.
generating a single instance, which is not  Is the term mode collapse is used as the 2D diffusion model generates multiple facet in single 3D generation author
- Method: While I believe the entropic regularization is somewhat new to the text-to-3D generation research, but it seems like it is just a classifier-free guidance training of LoRA parameters in VSD, which is already in part suggested from the Threestudio, which has an open-source re-implementation of ProlificDreamer.
- Evaluation: The paper only considers 4 prompts in its evaluation. I think the quantity of experimental validation does not suffice to support the author’s claim that ESD prevents Janus problem or mode collapse. I think the author should consider showing more experimental results to convince the readers.
Effect of ESD vs seed: As discussed in DreamFusion, the most simplest way to evade Janus' problem is to use different random seeds. Thus, if ESD is better at preventing Janus’ problem, the experiments should be designed to show how ESD successfully generates 3D content without Janus problem compared to ProlificDreamer or DreamFusion when repeated with different random seeds.
- Ablation study: The author provides brief qualitative examples on how the entropy regularization hyperparameter $\lambda$ affects the 3D generation. However, I think it would be much better how $\lambda$ affects the 3D generation throughout the generation (i.e., 3D optimization) process, or how it prevents the mode collapse issue. This is related to 1. that the quantification of mode collapse will strengthen the author’s point.

**Questions:**

Most of my concerns are listed in the Weakness part. Here, I will add some more questions regarding the paper:

- Could the author provide more details in their implementation? For example, which 3D backbones were used, or which diffusion models were used during generation? I think there is no clear description on the optimization.
- Does the author conduct an experiment on 3D mesh fine-tuning as done in ProlificDreamer? If so, is the proposed method (ESD) effective in 3D mesh optimization?
- Some minor questions; ain’t the $\partial g(\theta,c) / \partial \theta$ should be positioned at the right of the score difference term?

---

### Official Review · Reviewer_mdp1 · 2023-11-09

**Soundness:** 2 fair
**Presentation:** 2 fair
**Contribution:** 3 good
**Rating:** 5
**Confidence:** 2

**Summary:**

In this paper, the authors propose to mitigate the Janus artifact in Text-to-3D Generation. The Janus artifact, rooted in mode collapse, primarily arises from maximizing the likelihood of each dependent perspective. To tackle this issue, the authors introduce Entropic Score Distillation, supported by theoretical proof. Furthermore, the authors incorporate a classifier-free guidance trick to bolster the robustness of the proposed ESD method. The work is supported by both quantitative performance and visualization results.

**Strengths:**

- The manuscript introduces an innovative approach that offers a fresh perspective on mitigating the Janus artifact in text-to-3D generation, setting it apart from prior works.

- The manuscript includes detailed mathematical derivations and demonstrations supporting the proposed method.

**Weaknesses:**

In the experimental section, the authors demonstrate the model's effectiveness by presenting both qualitative and quantitative outcomes of 3D generation across various prompts. Nevertheless, these findings may not comprehensively represent the overall scenario, particularly in cases where the model's performance exhibits instability. It would enhance the study if additional analyses or discussions were included. These analyses could focus on identifying the specific target distribution (whether from a statistical or visualization perspective) where the baseline falls short or exhibits the janus artifact, and elucidate how the proposed model addresses and mitigates this issue.

For experimental results, the authors fall short in delivering a thorough analysis of the baseline model's performance based on the proposed metrics. Such an analysis is crucial to demonstrate the viability and relevance of the designed metrics. For insights into this aspect, please refer to Q1, Q2, and Q3.

**Questions:**

1. VSD appears more visually appealing than SDS; however, the discrepancy between this perception and the data presented in Table 1 requires elucidation. A discussion should be incorporated to shed light on the factors contributing to this observed contrast.

2. Despite the mathematical soundness of IQ and IV, their experimental justification is essential. A detailed analysis of SDS and VSD performance on IQ and IV metrics, accompanied by a relevant discussion, is suggested to strengthen their empirical foundation.

3. Section 6's definition of IQ and IV should include an explanation of the classifier employed and the rationale behind its selection.

4. The authors are encouraged to provide a brief discussion on the types of target distributions that are more susceptible to Janus artifacts, particularly from a statistical perspective (as depicted in Fig. 1). Additionally, an explanation of how the proposed methods effectively mitigate such artifacts would enhance clarity.

5. I would suggest the authors provide a rationale for not employing the four metrics utilized in Table 1 in the ablation study. A brief discussion on the reasons behind this choice and its implications for the quantitative evaluation of 3D results would be beneficial.